# Effect of Solution Temperature on Microstructure and Properties of Thixotropic Back-Extruded Tin–Bronze Shaft Sleeve

**DOI:** 10.3390/ma15155254

**Published:** 2022-07-29

**Authors:** Yuhang Zhou, Yunxin Cui, Qingbiao Zhang, Zhiqiang Yang, Yongkun Li, Han Xiao

**Affiliations:** Faculty of Materials Science and Engineering, Kunming University of Science and Technology, Kunming 650093, China; zhouyvhang@126.com (Y.Z.); cuiabao@126.com (Y.C.); zqb0325@163.com (Q.Z.); yangzhiqiang9991@163.com (Z.Y.); liyongkun@kust.edu.cn (Y.L.)

**Keywords:** semisolid forming, solution treatment, thixotropic back extrusion, microstructure, mechanical properties

## Abstract

To study the heat-treatment process of a semi-solid copper alloy, a thixotropic back-extruded tin–bronze shaft sleeve was heat-treated at 630 °C, 660 °C, 690 °C and 720 °C for 1 h, respectively. Microstructure changes and mechanical properties under different solution temperatures of shaft sleeve were characterized using a metallographic microscope (OM), scanning electron microscope (SEM), transmission electron microscope (TEM), hardness tester, and tensile tester. The results show that the tensile strength first increases and then decreases; the elongation decreases; and the Brinell hardness increases gradually with increasing solution temperature. When the solution treatment is at 690 °C, the tin–bronze shaft sleeve’s microstructure and comprehensive mechanical properties are the best. The shape factor is 0.75, the average grain size is 82.52 μm, the Brinell hardness is 122 HBW, the tensile strength is 437 MPa, and the elongation is 17.4%, which is 3.4 times higher than that before solution treatment.

## 1. Introduction

Tin–bronze, with its excellent wear resistance, corrosion resistance, and mechanical properties, has been widely used in the automotive, electronics, and aerospace industries, and in other fields [1]. For this high-melting-point alloy, traditional squeeze-casting greatly reduces the service life of the die due to the high pouring temperature, resulting in high production costs and limited application space. Metal semisolid-forming technology (Semisolid Metal Processing) has the characteristics of near-net forming and low forming temperature, which was first proposed by Flemings [2] and has been widely studied by scholars at home and abroad. Semisolid forming is mainly divided into rheological forming and thixoforming [3], in which thixoforming is considered to be more suitable for semisolid forming of high melting point alloys [4,5,6]. The microstructure uniformity has a great influence on the properties of thixoformed products [7,8]. In the thixoforming process, due to the difference in fluidity of solid–liquid two-phase, it is easy to cause solid–liquid phase separation and cause macro- and microsegregation of solid–liquid two-phase. Many scholars have conducted research on macrosegregation. By improving die structure through methods such as backward-extrusion, and by changing thermal process parameters, macrosegregation has been significantly improved [9,10]; however, it is still difficult to avoid microsegregation. Microsegregation also affects the mechanical properties of the alloy, especially for the tin–bronze alloy. This is due to the peritectic reaction of the tin–bronze alloy during non-equilibrium solidification and the formation of the δ phase (Cu_41_Sn_11_ phase) [11]. The content and distribution of the δ phase could affect the comprehensive mechanical properties of the alloy [12]. Park [13] and other researchers’ results show that too much δ phase in the tin–bronze alloy will reduce the strength of tin–bronze alloy; the hardness increases with increasing Sn content, and when the Sn content is 22 at%, the hardness almost reaches the upper limit. Li [14] showed that the elongation and tensile strength of the extruded parts formed via isothermal treatment of a tin–bronze semisolid slurry, prepared via a rheological forming process for 20 s, are increased; this may be due to the migration of the Sn element to the α-Cu matrix in the isothermal process, which makes the microstructure of the formed parts more uniform and the solution stronger. In the thixoforming process of tin–bronze, it is necessary to ensure a certain liquid phase rate to ensure good mold filling performance, which leads to the formation of more δ phases in the microstructure of the formed parts after solidification. These δ phases mainly exist in an intergranular structure, which affects the comprehensive mechanical properties of the alloy. At the same time, the comprehensive properties of the alloy are further reduced due to the existence of microsegregation, so it is necessary to solve this problem. In the actual production of metal parts, a reasonable heat-treatment process is often used to change the morphology, size, and distribution of phases in the structure, to further improve the mechanical properties of metal parts [15,16,17]. There have been many reports on the heat treatment of the Cu-Sn alloy. Mao [12] studied the microstructure evolution and mechanical properties of Cu-15Sn bronze samples before and after annealing. The results showed that the main strengthening mechanism of the parts after annealing changed from fine-grain strengthening to solution strengthening. Huang [18] et al. studied the annealing treatment of tin–bronze at 400 °C–800 °C for 1 h. It was found that when the annealing temperature increased to 500 °C, the microstructure recrystallized. The research on the heat-treatment process of semisolid-forming products is mostly focused on aluminum alloys and magnesium alloys [19,20], while the research on copper alloys is less abundant. In this paper, the effects of solution temperature on element distribution, microstructure uniformity, Brinell hardness, tensile strength, and the elongation of thixotropic back-extruded tin–bronze bushings were studied. The comprehensive mechanical properties of the alloy are further improved, which provides a reference value for the semisolid-forming process of tin–bronze.

## 2. Experimental Procedure

### 2.1. Material Preparation

The experimental material was cast ZCuSn10P1 copper alloy billet (25 × 25 × 60 mm). The solidus and liquidus temperatures of the alloy, measured using a synchronous thermal analyzer (STA449F3), were 837.2 ℃ and 1026.9 °C, respectively [14].

The thixotropic back-extruded processes were as follows: first, the as-cast billet was pre-annealed at 600 °C for 2 h and air-cooled to room temperature; then, it was unidirectionally cold-rolled to a thickness of 20 mm, and underwent isothermal treatment for 15 min at 900 °C. Next, it was transferred to the mold for back-extrusion forming, pressure holding for 10 s, and water quenching to make tin–bronze shaft sleeve parts. The extrusion pressure was 50 T, the extrusion speed was 15 mm/s, and the preheating temperature of the die was 400 °C. The shaft sleeves were treated via solution treatment at 630 °C, 660 °C, 690 °C, and 720 °C for 1 h, respectively, then quenched with water at room temperature. Figure 1a shows the schematic diagram of thixotropic back-extrusion process.

### 2.2. Microstructure Characterization

As shown in Figure 1b, metallographic samples were taken in the middle of the shaft sleeve, then polished using 400-, 600-, 800-, 1000-, and 1200-mesh sandpaper; then, they were mechanically polished using 1.0 μm diamond polishing paste, and finally, etched for about 3 s with 5% FeCl_3_ solution (5 mL FeCl_3_ + 10 mL HCl + 100 mL H_2_O). An optical microscope (Nikon MA200) was used to observe the metallographic structure of shaft sleeves at different solution temperatures and randomly produce five 100x metallographic diagrams. The Image-Pro Plus 6.0 software was used to calculate the intergranular structure content (*K*), average grain size ($D_{eq}$), and shape factor (*R*) [21,22] using Formulas (1)–(3).
(1)$$K=1-\frac{A_{N}}{A}$$
(2)$$D_{eq}=\frac{\sum_{N=1}^{N}\sqrt{4A_{N}/\pi }}{N}$$
(3)$$R=\frac{\sum_{N=1}^{N}\frac{4\pi A_{N}}{C_{N}^{2}}}{N}$$

Here, *A_N_*, *N*, and *C_N_* represent the area, quantity, and perimeter of grains, respectively. *A* represents the total area. The closer the shape factor *R* is to 1, the better the grain roundness.

The microstructure and energy-dispersive X-ray spectroscopy (EDXS) analyses of the shaft sleeves were carried out via scanning electron microscope (ZEISSEVO18, acceleration voltage: 15 KV, signal: SE) and its EDXS detector (acceleration voltage: 15 KV). The phases were identified via X-ray diffraction (Empyrean, scan rate: 10°/min, wavelength: 1.54184 Å) and transmission electron microscope (TEM, Talos 200, HAADF-STEM, acceleration voltage: 200 KV, sample preparation method: FIB), and the phase distribution characteristics were observed.

### 2.3. Mechanical Property Experiment

Three positions were randomly selected on the metallographic specimen, and the hardness of the shaft sleeve was tested using an HBE-3000A electronic Brinell hardness tester. The test conditions were as follows: indenter diameter: 10 mm, holding time: 30 s, load: 62.5 N. The average value was taken as the final result. A uniaxial tensile test of the sleeve specimen at room temperature was carried out using a CMT300 tester at a tensile rate of 2 mm/min. Three tensile samples were tested at each solution temperature, and the average value was taken as the final result. The sampling position is shown in Figure 1b.

## 3. Results and Discussion

### 3.1. Effect of Solution Temperature on Microstructure of Tin–Bronze Shaft Sleeve

According to the phase diagram of the Cu-Sn binary alloy [23,24,25], when the solution temperature is between 630 °C and 720 °C, the solid solubility of the Sn element in the α-Cu matrix decreases gradually with increasing temperature, while the theoretical solid solubility values are 9.01, 8.79, 8.40, and 8.14 at%, respectively.

The metallographic structure of the tin–bronze shaft sleeve after solution treatment is shown in Figure 2. The intergranular structure content of the tin–bronze shaft sleeve before and after solution treatment is shown in Figure 3. As shown in Figure 2a, before solution treatment, there are more intergranular structures and there is more segregation (shown by dotted lines in the figure), and the grain roundness is poor; this has an adverse effect on the comprehensive mechanical properties of the shaft sleeve. The intergranular structure is mainly α + δ + Cu_3_P eutectic phase.

As shown in Figure 2b,c, when the solution temperature is 630 °C, the grains coarsen considerably and the morphology of the semisolid spherical grains disappears.

At this temperature, the diffusion rate of Sn into the grains is faster, the decomposition of the intergranular Cu_41_Sn_11_ phase is faster in the process of the solution treatment, and the hindrance effect of the intergranular structure on grain growth is greatly weakened, which leads to abnormal grain growth. In addition, the enrichment region of the Sn element appears at the grain boundary, which may be due to the higher solid solubility of the Sn element in the α-Cu matrix; moreover, the diffusion rate of the Sn element from intercrystal to intracrystal is faster compared with the diffusion rate of the Sn element within the grain.

As shown in Figure 2d, when the solution temperature is 660 °C, the area rich in Sn is reduced compared with that at 630 °C. This is due to the decrease in the solid solubility of the Sn element in the α-Cu matrix, the diffusion rate of Sn into the matrix, and the difference between the diffusion rate and the rate of re-diffusion to the center. The grains are still coarse at this temperature, which is because the intergranular structure decomposes so much that it is easy for the grains to merge and grow in the process of growth, which eventually leads to the destruction of the semisolid spherical structure.

As shown in Figure 2e, when the solution temperature is 690 °C, the solid solubility of the Sn element in the α-Cu matrix is lower, the diffusion rate of the Sn element decreases further, and the intergranular structure decomposes slowly, which makes it difficult for the grains to merge and grow. Under the pinning effect of the intergranular structure on the grain boundary, the grains can grow more uniformly. When the Sn element diffuses into the grains, it is easier for it to spread and distribute uniformly among the grains, and the intergranular structure is more evenly distributed after water quenching. Finally, a better semisolid microstructure is obtained.

As shown in Figure 2f, when the solution temperature is 720 °C, the microstructure morphology is close to 690 °C, but the difference in the grain size of the structure is larger and more small grains can be seen around the large grains. This is because large grains grow and small grains decrease in the process of grain growth. When the solution temperature is higher, the grain growth rate is faster, resulting in non-uniform grain size.

The grain size and shape factor of the tin–bronze shaft sleeve before and after solution treatment are shown in Figure 4. Compared with that before solution treatment, when the solution temperature is 690 °C, the grain size increases from 68.42 μm to 82.52 μm, and the shape factor increases from 0.65 to 0.75. This is because under the action of kinetics, the grain grows obviously during solution treatment, the zigzag boundary between the original grain and the intergranular structure decreases, and the grain boundary is smoother, so the shape factor increases slightly. When the solution temperature is 720 °C, the grain size increases to 85.87 μm and the shape factor is 0.74. This is because when the solution temperature is higher and the holding time is the same, the grain growth rate is faster, resulting in a slight increase in grain size and little change in the shape factor. From the above analysis, the grain size and roundness of tin–bronze are best when the solution temperature is 690 °C.

### 3.2. Evolution of Element Distribution and Phase Composition of Tin–bronze Shaft Sleeve

The element distribution, determined via EDXS mapping measurements of the tin–bronze shaft sleeve before and after solution treatment, are shown in Figure 5. The distribution map of Sn elements shows that Sn elements are enriched between grains before solution treatment, and there is an obvious segregation phenomenon; meanwhile, after solution treatment, the Sn elements tend to be uniformly distributed, and the segregation is improved; this may be beneficial to improve the mechanical properties of the parts. This is because the solution treatment makes the Sn elements in the intergranular structure diffuse into the α-Cu matrix, thus improving the segregation of Sn elements.

Point-scanning quantitative analysis of the center of several randomly selected α-Cu grains was conducted, and the results of the average content of Sn elements are shown in Table 1. The average content of Sn in the α-Cu matrix after different temperatures of solution treatment is significantly higher than that before solution treatment, and decreases gradually with increasing solution temperature, which is consistent with the change in solid solubility of Sn in the α-Cu matrix.

The distribution map of the P element shows that it is enriched in the intergranular structure before solution treatment. When the solution temperature is 630 °C and 660 °C, the distribution of the P element in the matrix is relatively uniform and there is slight enrichment of the P element, but it is not obvious. The enrichment of the P element between crystals was observed at solution temperatures of 690 °C and 720 °C. This is because the miscibility of the P element and Cu element is poor, but the binding ability of Cu-P is very high; therefore, it is difficult to decompose Cu_3_P via solution treatment, and it is easy to form a stable compound Cu_3_P [24] where the P element is segregated. Not only does the addition of the P element give tin–bronze good deoxidization ability during casting, but the formed Cu_3_P can also increase the wear resistance of the material [25].

The XRD patterns of tin–bronze shaft sleeves before and after solution treatment are shown in Figure 6. It can be seen in Figure 6 that the microstructure before solution treatment is composed of α-Cu, Cu_41_Sn_11_, Cu_3_P, and Cu_13.7_Sn, and after solution treatment, the microstructure is composed of α-Cu, Cu_41_Sn_11_, and Cu_3_P. Metastable-phase Cu_13.7_Sn is derived from the martensitic transformation of peritectic-phase Cu_5_Sn. The formation of the Cu_13.7_Sn phase may be due to the higher cooling rate during water quenching. After solution treatment, there is no obvious Cu_41_Sn_11_ diffraction peak, but it combines with the α-Cu peak to form a single peak. This is due to the diffusion of Sn atoms into the α-Cu matrix and the decomposition of the Cu_41_Sn_11_ phase in the intergranular structure, resulting in a decrease in the Cu_41_Sn_11_ diffraction-peak intensity. At the same time, due to the solution of Sn atoms in the α-Cu matrix, lattice distortion is caused and the α-Cu diffraction peak shifts to a small angle, so the double peaks merge into a single peak. In addition, there is a Cu_3_P peak after solution treatment, indicating that the microstructure of the thixotropic reverse-extrusion sleeve after solution treatment is close to the equilibrium phase of the ZCuSn10P1 copper alloy, namely α-Cu with an FCC structure and Cu_3_P with a hexagonal structure [26]. By comparing the positions of XRD diffraction peaks at different solution temperatures, longitudinally, in Figure 6b, it is found that with decreasing solution temperature, the α-Cu diffraction peak shifts to the left. This is mainly due to the decrease in solution temperature and the increase in Sn content in the α-Cu matrix, resulting in greater lattice distortion. When the solution temperature is 660 °C, the diffraction peak obviously shifts to the left, which is due to more Sn elements being dissolved, which leads to more serious lattice distortion. The Cu_13.7_Sn diffraction peak disappears after solution treatment, which may be due to the decrease in intergranular Sn element content and the low content of the Cu_13.7_Sn phase formed after water quenching.

As shown in Figure 7, the microstructure of the tin–bronze shaft sleeve after solution treatment at 690 °C for 1 h was analyzed via TEM. From the energy spectrum (Figure 7b,c) after surface scanning, it can be seen that the P element is enriched in the intercrystalline space, while the Sn element is only enriched at the edge of the grain boundary. This is because in the process of solution treatment, the Sn elements enriched between grains first migrate to the grain boundary, and then, gradually spread into the grain; thus, the enrichment phenomenon of Sn is formed at the edge of the grain boundary after solution treatment. The results of the dot-scan analysis in different regions (Figure 7d) are shown in Table 2. The results show that there is still an α-Cu phase in the intergranular structure. An SAED analysis of regions A and area B was carried out, and the diffraction pattern and calibration are shown in Figure 7e,f. The calibrated phases are the Cu_41_Sn_11_ and Cu_3_P phases, and their zone axes are [01$\bar{1}$] and [001], respectively. The lattice constant of the Cu_41_Sn_11_ phase (FCC) is a = 1.797 nm, and the lattice constant of nm, the Cu_3_P phase (HCP), is a = b = 0.708 nm, c = 0.715 nm. The intergranular structure should be a eutectoid structure composed of the Cu_41_Sn_11_ and Cu_3_P phases, and a small amount of the α-Cu phase (α-Cu + Cu_41_Sn_11_ + Cu_3_P). The existence of the Cu_13.7_Sn phase was not found using TEM, which indicates that the content of the Cu_13.7_Sn phase was low after solution treatment, and the effect on the semisolid’s microstructure and properties was negligible.

### 3.3. Microstructure Evolution Mechanism of Tin–bronze Shaft Sleeve during Solution Treatment

Combined with the results of the microstructure morphology and phase analysis of semisolid back-extruded tin–bronze shaft sleeves before and after solution treatment, the main mechanisms of microstructure evolution are grain coalescence and the ripening mechanism [27,28]. Figure 8 is a schematic diagram of the microstructure evolution mechanism of the semisolid back-extruded copper alloy during solution treatment. In the initial stage of the solution process, the intergranular eutectoid structure is decomposed, and the Sn elements are first enriched at the grain boundaries and sharp corners; this results in the passivation of the sharp corners of the grains and their gradual transition to a round shape, and the irregular parts of the edges of the large grains will break and form small grains, which are distributed around the large grains [29]. As the Sn element gradually diffuses into the α-Cu matrix, it also diffuses between the grains (as shown by the red arrow). At the same time, under the driving force of the decrease in interfacial energy, the grains will grow continuously, which is mainly characterized by the continuous growth of large grains and the decrease in small grains.

When the solution temperature is low and the solid solubility of the Sn element is high, the diffusion rate of the Sn element is faster, and it is easier for the grains to contact each other where the intergranular structure is thinner; this increases the probability of grain merging and growth, and hinders the migration of Sn element between grains. With the continuous decrease in the intergranular structure of Sn element diffusion, the limiting effect of intergranular structure on grain growth is greatly weakened and inhomogeneous. The small grains in the microstructure gradually decrease or even disappear, and the large grains continue to grow and be elongated; this results in the coarse and irregular shape of the grains, which cannot maintain their semisolid near-spherical morphology.

When the solution temperature is high and the solid solubility of the Sn element is low, due to the slow diffusion rate of Sn into the matrix, the Sn element is more likely to spread and distribute among grains, which leads to a slower decomposition rate of intergranular structure and a lower probability of merging and growing between grains. The essence of grain growth is the migration of the grain boundary, and the intergranular structure has a pinning effect on the grain boundary, so it can hinder grain growth. When the solid solubility is high, the distribution of the intergranular eutectoid structure is more uniform, and the intergranular structure hinders grain growth more strongly and uniformly (as shown by the black arrow); this results in a smaller grain size and better roundness, and the microstructure maintains its spherical grain morphology.

### 3.4. Effect of Solution Temperature on Brinell Hardness of Tin–Bronze Shaft Sleeve

The average Brinell hardness of the tin–bronze shaft sleeve before and after solution treatment is shown in Figure 9. As can be seen in Figure 9, the Brinell hardness increases continuously with increasing solution temperature. When the solution temperature increases from 630 °C to 720 °C, the Brinell hardness increases from 118 HBW to 123 HBW, an increase of 4.2%. This is due to the fact that with increasing solution temperature, the solid solubility of the Sn element in the α-Cu matrix decreases, the diffusion of the Sn element into the α-Cu matrix decreases, and more Cu_41_Sn_11_ phases are uniformly distributed between grains. Cu_41_Sn_11_ has high brittleness [30]; therefore, the average Brinell hardness increases with increasing solution temperature, and the average Brinell hardness is 122 HBW at 690 °C. The average Brinell hardness of the thixotropic reverse-extrusion sleeve before solution treatment is 128 HBW; this shows that the average Brinell hardness is lower overall after solution treatment, which is related to the decrease in the Cu_41_Sn_11_ phase after solution treatment.

### 3.5. Effect of Solution Temperature on Tensile Properties of the Tin–Bronze Shaft Sleeve

The tensile strength and elongation of tin–bronze shaft sleeves before and after solution treatment are shown in Figure 10. After solution treatment, the tensile strength and elongation of the shaft sleeves are mainly affected by solution strengthening, fine-grain strengthening, intergranular structure content, and microstructure homogenization [16]. The δ phase, which mainly exists in the intergranular structure, is beneficial to the improvement of tensile strength, but there are some restrictions on its plasticity. Before solution treatment, the tensile strength of the shaft sleeve is 400 MPa and the elongation is 5.1%.

When the solution temperature is 630 °C, the elongation is greatly improved. This is because, on the one hand, the δ phase decomposes more and the intergranular structure decreases obviously; on the other hand, the semisolid microstructure disappears and the fine-grain strengthening weakens, so the elongation is greatly improved. However, under the action of solid-solution strengthening, the tensile strength changes little.

When the solution temperature is 660 °C, there is little difference in the microstructure, tensile strength, and elongation compared with 630 °C. The properties of the alloy are mainly affected by the content of intergranular structure. Because the solid solubility is relatively low and the decomposition of the δ phase is lower, the tensile strength is slightly higher and the elongation is slightly lower.

When the solution temperature is 690 °C, at the same time as solution strengthening, more of the intergranular structure is retained and evenly distributed; additionally, the microstructure still has semisolid spherical grain morphology, so the tensile strength is greatly improved. At the same time, with the obvious increase in intergranular structure, the elongation decreases significantly, but it still meets the requirements of a tin–bronze shaft sleeve.

When the solution temperature is 720 °C, the content of intergranular structure increases to 5.37%, which is the main reason for the further decrease in tensile strength and elongation.

The tensile-fracture morphology of the tin–bronze shaft sleeve before and after the solution is shown in Figure 11. There is an obvious dissociation platform on the tensile-fracture surface before solution treatment, indicating that the elongation is poor and the fracture mode should be a dissociation fracture. When the solution temperature is 630 °C, a large number of dimples appear on the tensile-fracture surface, from which it can be judged that the fracture mode is a ductile fracture with high elongation. When the solution temperature is 660 °C, the existence of dimples and a small number of second-phase particles can be seen. When the solution temperature is 690 °C, more second-phase particles appear in the structure, and the second-phase strengthening improves the material’s tensile strength. At the same time, a large number of dimples can be seen, indicating that the plasticity of the material is good. This is because the semisolid spherical-grain α-Cu solution occupies the main microstructure, but the intergranular structure is lower; additionally, the α-Cu solution has good plastic deformation ability. When subjected to external forces, α-Cu undertakes most of the deformation.

When the solution temperature is 720 °C, there are tearing corrugations, and there are fewer dimples than other fracture morphologies; this may be due to the further coarsening of grains, which increases the brittleness of the material. Moreover, with increasing solution temperature, the increase in the intergranular structure also increases the risk of brittle fracture, and the fracture mode is the transition from ductile fracture to brittle fracture.

## 4. Conclusions

(1) In the process of solution treatment, the Sn element will diffuse into the α-Cu matrix, which mainly comes from the decomposition of the intergranular δ phase.

(2) The microstructure of the tin–bronze shaft sleeve after solution treatment is mainly affected by the solid solubility of the Sn element in the α-Cu matrix. When the solid solubility is low, the microstructure can maintain a good semisolid spherical grain morphology. The microstructure morphology is best after solution treatment at 690 °C for 1 h.

(3) The content and distribution of the intergranular structure have a significant effect on the mechanical properties of semisolid tin–bronze alloy. The intergranular structure is composed of the α-Cu, δ, and Cu_3_P phases, and the δ and Cu_3_P phases are calibrated. The crystal band axes are [011] and [001], respectively, and the lattice constants are 1.797 nm and a = b = 0.708 nm, c = 0.715 nm, respectively.

(4) After solution treatment at 690 °C for 1 h, the comprehensive mechanical properties are at their best. The Brinell hardness is slightly lower than before solution treatment, the tensile strength is guaranteed and slightly increased under the action of solution strengthening and second phase strengthening, and the elongation is increased 3.4 times. This is mainly due to the decrease in and uniform distribution of the intergranular structure.

## Figures and Tables

**Figure 1 materials-15-05254-f001:**
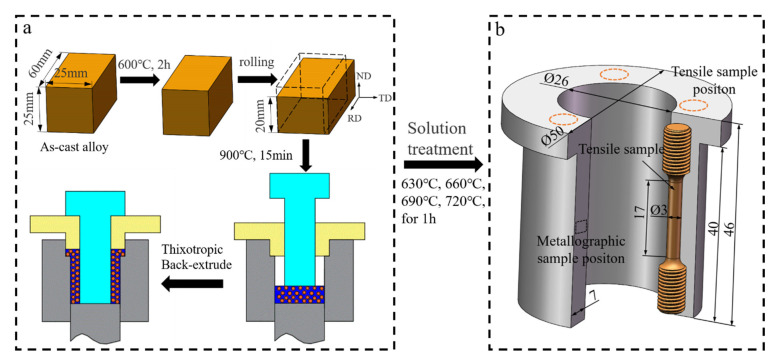
Thixotropic back-extrusion process schematic diagram (**a**) and part drawing (**b**).

**Figure 2 materials-15-05254-f002:**
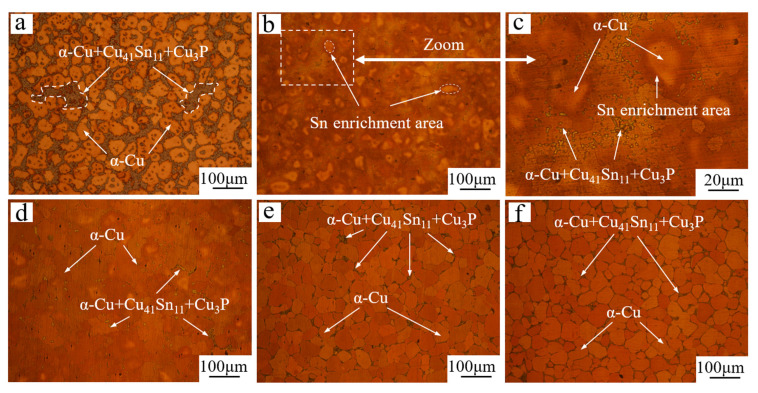
Optical micrographs (bright field contrast) of thixotropic back-extruded tin–bronze shaft sleeve: (**a**) before solution, (**b**) 630 °C, (**c**) partial magnification of Figure b, (**d**) 660 °C, (**e**) 690 °C, and (**f**) 720 °C.

**Figure 3 materials-15-05254-f003:**
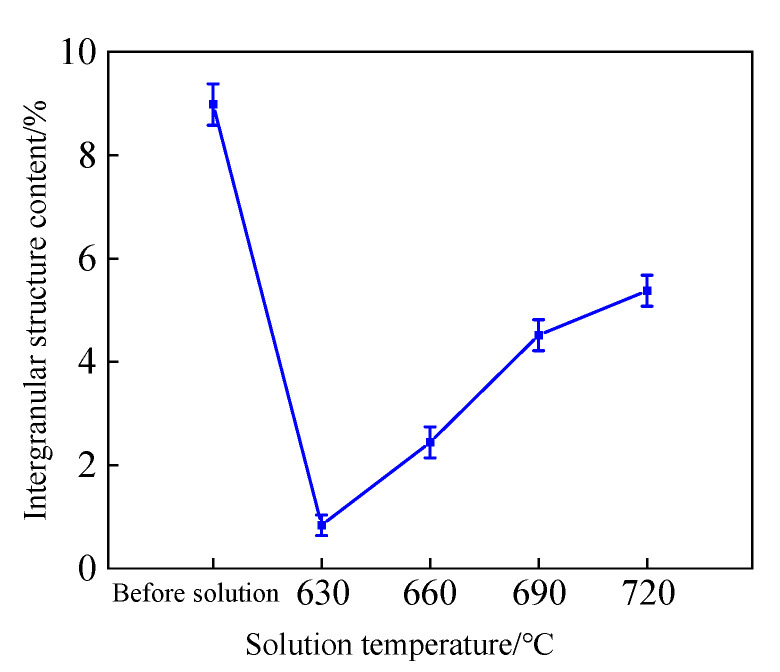
Intergranular structure content of tin–bronze shaft sleeve.

**Figure 4 materials-15-05254-f004:**
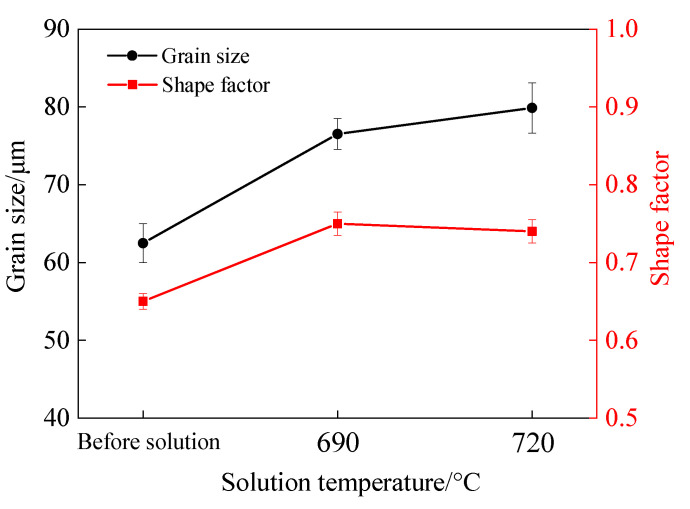
Grain size and shape factor of tin–bronze shaft sleeve.

**Figure 5 materials-15-05254-f005:**
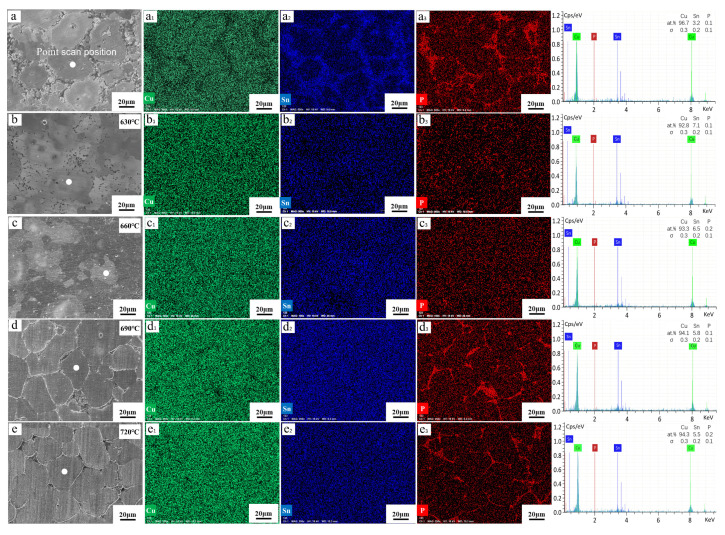
SEM images and EDS element distribution of tin–bronze shaft sleeve before and after solution treatment: (**a**–**a_3_**) before solution, (**b**–**b_3_**) 630 °C, (**c**–**c_3_**) 660 °C, (**d**–**d_3_**) 690 °C, and (**e**–**e_3_**) 720 °C.

**Figure 6 materials-15-05254-f006:**
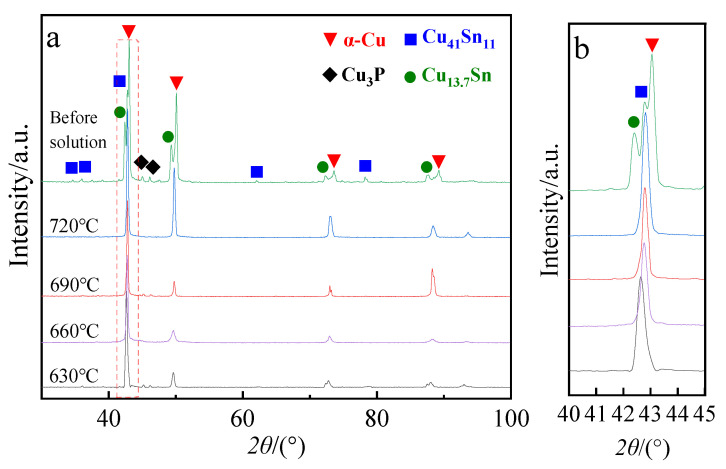
XRD pattern of the (**a**) tin–bronze shaft sleeves at different solution temperatures and (**b**) magnified local diagram.

**Figure 7 materials-15-05254-f007:**
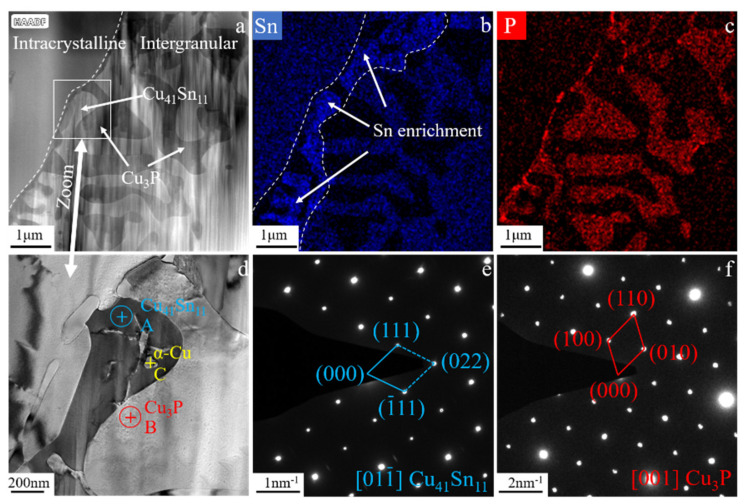
Bright-field TEM images of tin–bronze shaft sleeve after solution treatment at 690 °C/1 h (**a**); EDS diagram of Sn (**b**) and P (**c**) element; partial magnification of Figure a (**d**); and selected-area electron diffraction (SAED) patterns of regions A (**e**) and B (**f**) in Figure d.

**Figure 8 materials-15-05254-f008:**
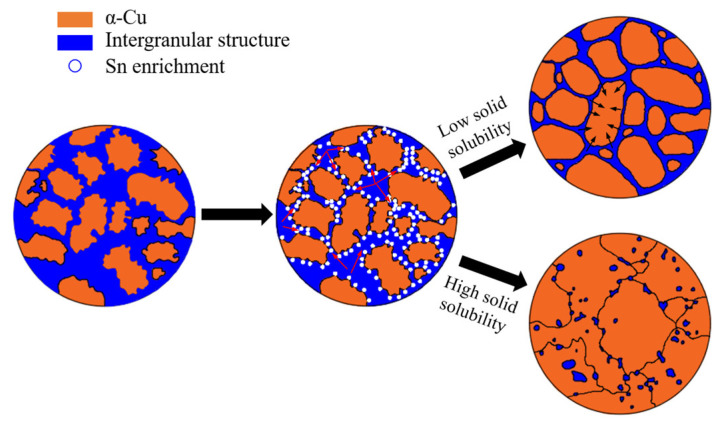
Schematic diagram of microstructure evolution mechanism during the solution of the thixotropic back-extruded tin–bronze shaft sleeve.

**Figure 9 materials-15-05254-f009:**
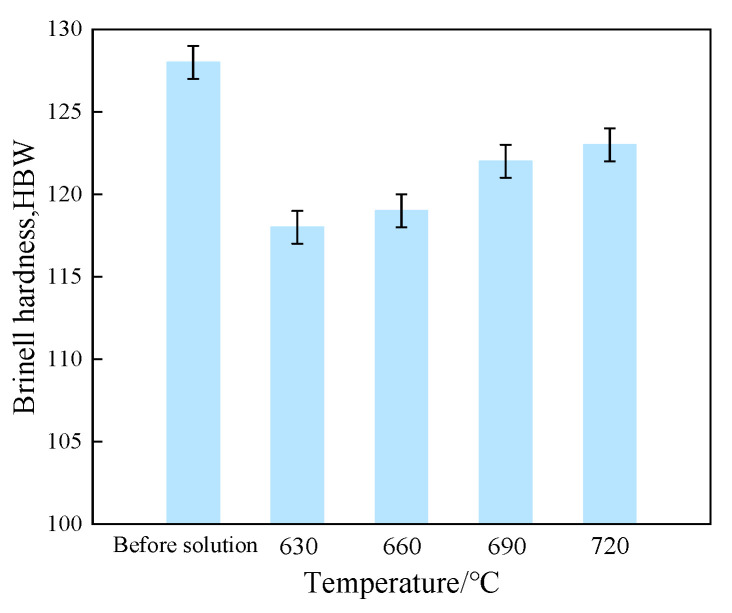
Average Brinell hardness of tin–bronze shaft sleeves at different solution temperatures.

**Figure 10 materials-15-05254-f010:**
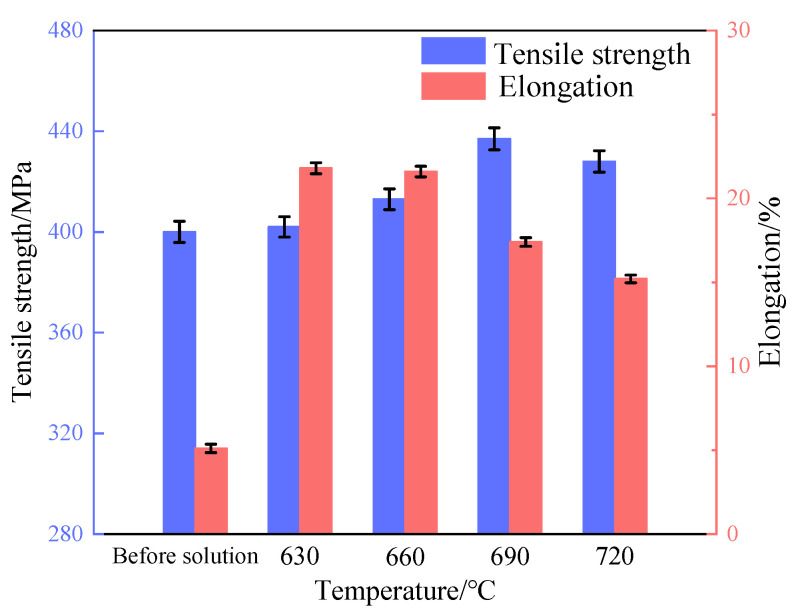
Tensile strength and elongation of tin–bronze shaft sleeves at different solution temperatures.

**Figure 11 materials-15-05254-f011:**
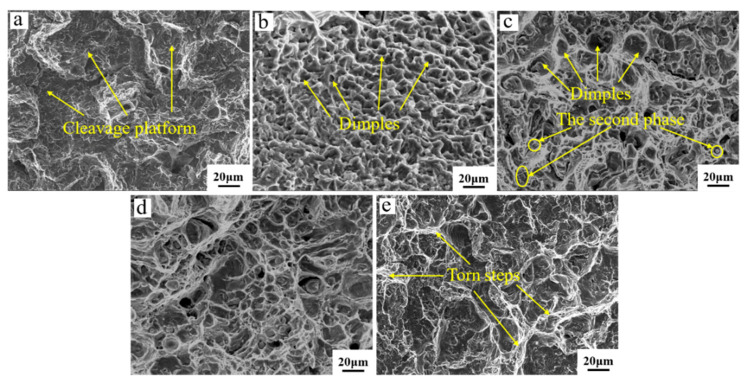
Tensile-fracture morphology of tin–bronze shaft sleeves at different solution temperatures: (**a**) before solution, (**b**) 630 °C, (**c**) 660 °C, (**d**) 690 °C, and (**e**) 720 °C.

**Table 1 materials-15-05254-t001:** Average fraction of Sn element in the α-Cu matrix.

Solution Temperature (°C)	Before Solution	630	660	690	720
Sn(at)%	3.2	7.1	6.5	5.8	5.5

**Table 2 materials-15-05254-t002:** Fraction of Cu, Sn, and P elements from point scanning in different regions.

Element	Cu	Sn	P
A (at)%	85.71	13.99	0.30
B (at)%	81.28	0.16	18.56
C (at)%	95.40	3.89	0.71

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
