# Peer review of "Effect of Solution Temperature on Microstructure and Properties of Thixotropic Back-Extruded Tin–Bronze Shaft Sleeve"

_materials, 2022, doi:10.3390/ma15155254_

Round 1
Reviewer 1 Report
In the review of the manuscript titled: Effect of solution temperature on Microstructure and Properties of Thixotropic Back-extruded Tin Bronze Shaft sleeve. The authors have provided the good description and methodology is also fine. I would like to see this article publish but after some minor modifications as follow;
1. Why the authors only selected these values of temperatures 630 ℃, 660 ℃, 690 ℃ and 720 ℃?
2. The authors stated that the tensile strength increases firstly and then decreases, the elongation decreases, and the Brinell hardness increases gradually with the increase of solution temperature. The authors are requested to add some discussion to support their claim.
3. The authors did not give any information about the Pro-Plus software in this paper. They are suggested to add few lines to explain image Pro-Plus software used to calculate the intergranular structure content.
4. Why the diffusion rate of the Sn element is faster when the solution temperature is low and the solid solubility of the Sn element is high?
5. How the coarsening of grains, which increases the brittleness of the material?
Reviewer 2 Report
The manuscript „Effect of solution temperature on Microstructure and Properties of Thixotropic Back-extruded Tin Bronze Shaft sleeve “ by Yuhang Zhou et al. is suitable for publication with minor revision.
